# A Pilot Randomised Clinical Trial of a Novel Approach to Reduce Sedentary Behaviour in Care Home Residents: Feasibility and Preliminary Effects of the GET READY Study

**DOI:** 10.3390/ijerph17082866

**Published:** 2020-04-21

**Authors:** Maria Giné-Garriga, Philippa M. Dall, Marlene Sandlund, Javier Jerez-Roig, Sebastien F. M. Chastin, Dawn A. Skelton

**Affiliations:** 1School of Health and Life Sciences, Glasgow Caledonian University, Cowcaddens Road, Glasgow G4 0BA, UK; Philippa.Dall@gcu.ac.uk (P.M.D.); Sebastien.Chastin@gcu.ac.uk (S.F.M.C.); Dawn.Skelton@gcu.ac.uk (D.A.S.); 2Department of Physical Activity and Sport Sciences, Faculty of Psychology, Education and Sport Sciences (FPCEE) Blanquerna, Ramon Llull University, Císter 34, 08022 Barcelona, Spain; 3Department of Community Medicine and Rehabilitation, Umeå University, 901 87 Umeå, Sweden; marlene.sandlund@umu.se; 4Research Group on Methodology, Methods, Models and Outcome of Health and Social Sciences (M3O), Faculty of Health Sciences and Welfare, University of Vic-Central University of Catalonia (UVIC-UCC), 08500 Vic, Spain; javier.jerez@uvic.cat; 5Department of Movement and Sport Science, Ghent University, St. Pietersnieuwstraat 33, 9000 Ghent, Belgium

**Keywords:** care home residents, sedentary behaviour, co-creation, feasibility, acceptability

## Abstract

Care-home residents are among the most sedentary and least active of the population. We aimed to assess the feasibility, acceptability, safety, and preliminary effects of an intervention to reduce sedentary behaviour (SB) co-created with care home residents, staff, family members, and policymakers within a pilot two-armed pragmatic cluster randomized clinical trial (RCT). Four care homes from two European countries participated, and were randomly assigned to control (usual care, CG) or the Get Ready intervention (GR), delivered by a staff champion one-to-one with the care home resident and a family member. A total of thirty-one residents participated (51.6% female, 82.9 (13.6) years old). GR involves six face to face sessions over a 12-week period with goal-oriented prompts for movement throughout. The feasibility and acceptability of the intervention were assessed and adverse events (AEs) were collected. The preliminary effects of the GR on SB, quality of life, fear of falling, and physical function were assessed. Means and standard deviations are presented, with the mean change from baseline to post-intervention calculated along with 95% confidence intervals. The CG smoked more, sat more, and had more functional movement difficulties than the GR at baseline. The GR intervention was feasible and acceptable to residents and staff. No AEs occurred during the intervention. GR participants showed a decrease in daily hours spent sitting/lying (Cohen’s d = 0.36) and an increase in daily hours stepping, and improvements in health-related quality of life, fear of falling, and habitual gait speed compared to usual care, but these effects need confirmation in a definitive RCT. The co-created GR was shown to be feasible and acceptable, with no AEs.

## 1. Introduction

Improvements in public health care and advances in medical science have significantly extended life expectancy, and projections show continued increases in longevity. The number of older adults will increase in the coming decades at a faster pace than any other age segment of the European Union’s population, to reach 22% of the world total population [1,2]. Accompanying advancing age, risk of physical and cognitive decline, chronic diseases and comorbidities are more frequent [3]. Risk of institutionalization also increases with age, and one in four older adults will spend time in a care home in the United Kingdom [4]. Care-home residents are among the frailest of the population because of their physical dependency [5], cognitive impairment [6], multimorbidity, and polypharmacy [5]. 

A recent report of the European Commission (EC) concluded that, between 2016 and 2070, public expenditure related to ageing could increase from 1.6% to 26.7% of gross disposable income due to the growth of social and health costs [7]. Health-based research in this vulnerable population is necessary to try and address the challenges of their health care and ensure that robust, evidence-based service improvements are developed and implemented [8]. Compared with ageing research overall, research in long-term care remains relatively underdeveloped [9]. However, the recruitment of vulnerable older adults to research has reported low refusal rates, suggesting their willingness to be involved when given the opportunity [10,11]. 

Even though increased physical activity (PA) and reduced sedentary behaviour (SB) had been associated with numerous health-related benefits [12,13,14,15,16,17,18,19,20,21,22], there has been a lack of studies focused on reducing SB and enhancing movement (as opposed to specific exercise interventions) in institutionalized older adults [23,24]. Older adults are the most sedentary of any age group [25], with care home residents being the most sedentary older adults [26]. Older adults are also insufficiently active with only about 11% meeting the current PA recommendations [27], creating an environment ripe for movement behaviour intervention research. 

In the scientific literature, we find numerous interventions aimed at increasing levels of PA and reducing SB of older adults that have achieved limited success particularly over the long term and when implemented in real-life conditions [16,28,29,30,31,32,33]. It is widely recognised that there is a huge gap between the development of evidence-based interventions for public health and health promotion and their successful and sustainable implementation. The development of alternative approaches based on end-users’ preferences and which implement behavioural change concepts have been repeatedly requested to enhance sustainable changes [34,35]. A recent study reported that, according to care home staff members, residents like to be involved in most decision-making regarding the activities they are offered, and the involvement of the resident’s close relatives was seen as essential if residents were to be supported to be engaged [36]. Also, a recent systematic review showed that care-home residents could be successfully involved in participatory research [37]. To our knowledge, there is no previous experience involving care home residents, staff, and family members to co-create an individualized intervention to reduce SB and enhance movement throughout the day. 

Thus, within a pilot two-armed pragmatic cluster randomized clinical trial (RCT), we aimed to assess the feasibility, acceptability, and safety of an intervention to reduce SB and enhance PA, which was previously co-created with care home residents, staff, family members, and policymakers [38]. Our secondary aim was to assess the preliminary effects of the intervention to reduce SB and improve health-related outcomes. We hypothesized that the intervention would be feasible and well accepted by the participants and staff members.

## 2. Methods

### 2.1. Study Design

The protocol of the GET READY study was registered in ClinicalTrials.gov, with the identifier NCT03505385, and the study protocol had been previously published [39]. An earlier stage of the GET READY project integrated a service-learning methodology into physical therapy and sport sciences university degrees by offering students individual service opportunities with residential care homes. They were tasked to co-create the best suited intervention to reduce the SB of residents and enhance movement throughout the day, together with researchers, end-users, care staff members, family members, and policymakers [38]. In the current stage of the GET READY project, we conducted a pilot two-armed pragmatic cluster randomized clinical trial with baseline and end-of-intervention assessments.

The study was approved by the Scotland A Research Ethics Committee (Glasgow, 19/SS/0017), and the Faculty of Psychology, Education and Sport Sciences Blanquerna Ethics Committee (Barcelona, 026/2018). All participants signed an informed consent prior to participation and if a resident was screened as having mild to moderate dementia with the six-item screener to identify cognitive impairment [40], the personal legal representative was also asked to sign the informed consent in his/her behalf.

### 2.2. Participants

Five care homes were initially recruited (3 in Glasgow and 2 in Barcelona) and agreed to participate in the study. A care home was defined as a long-term care setting where people live and have their care needs met in homely surroundings, usually for people needing more care than they could get in their own home or in supported housing. We recruited public care homes both large (≥100 beds) and medium-sized (40–99 beds) in each county. The study was presented to staff members and managers from each setting and a total of six Staff Champions in Glasgow and five in Barcelona were recruited. All staff champions were trained with the first part of the intervention manual focused on the study aims, recruitment procedures, and how to gain informed consent from participants. The second part of the intervention manual related to the intervention components and activities of each session, and its training was only delivered to the staff champions of the intervention sites. Sites were randomly assigned to control or intervention after the residents’ baseline assessments, using opaque and sealed envelopes. Unfortunately, one care home in Glasgow (control site) had to be withdrawn from the study due to lack of consistent contact with the managers. Thus, we ran the study with two intervention sites in Glasgow, one intervention site in Barcelona, and one control site in Barcelona. 

Staff champions were asked to recruit residents of both sexes willing to participate. There were two exclusion criteria: residents with an end-stage disease, and/or with severe dementia. One family member of each participating resident was asked to join the study. Recruitment ran for 6 months from April to September 2019 and the staff champions were in charge of inviting all care home residents who met the inclusion criteria. A total of 31 residents, 9 from Glasgow and 22 from Barcelona (Table 1), from 4 care homes participated in the study. The mean age (SD) of participants in the GR and CG groups were 83.2 (9.1) and 82.7 (13.2) years old, respectively.

### 2.3. Outcomes

Baseline assessments were blinded to group allocation, and post-intervention assessments were performed by the same researcher who was at that point not blind to the site allocation. Staff champions at the care homes provided clinical and demographic information to allow description of the care home residents (Table 1).

To assess the feasibility of the study protocol, effectiveness of methods of recruitment, and enrolment, retention, and attendance rates with respect to the face-to-face behavioural sessions, and methods of data collection were assessed. To assess the acceptability of the intervention, all participants (with the personal legal representative if appropriate) were asked to rate their level of satisfaction with (a) the assessments and (b) the intervention using a 5-point Likert-type scale. We collected information about adverse events (e.g., safety issues) and other intervention-related issues in a monthly meeting/telephone call with the Staff Champions. 

To assess the preliminary effects of the GET READY intervention (GR), each participant (accompanied by the personal legal representative if necessary) was individually interviewed and the following outcomes were assessed at both time points (baseline and post-intervention): (a) self-rated health and health-related quality of life with the EUROQoL-5D questionnaire [41]; (b) sedentary behaviour with the sedentary behaviour questionnaire [42]; (c) fear of falling with the modified falls efficacy scale - international (MFES) [43]; (d) activities of daily living with the Katz scale [44]; and (e) functional performance with the short physical performance battery (SPPB), that includes balance in side-by-side, semi-tandem and tandem positions, habitual gait speed in a 4-meter course, and time to perform 5 chair stands [45]. 

The care home residents were asked to wear an ActivPAL3^TM^ monitor (PAL Technologies, Glasgow, UK), a valid “gold standard” method to measure SB [46,47]. The activity monitor was waterproofed (nitrile sleeve), secured onto the right anterior mid-line of the right thigh with a hypoallergenic patch (PAL stickie), and covered with a waterproof dressing (Opsite Flexifix). The monitor was worn for 24 h per day over seven days continuously, and participants recorded in a sleep diary their awake/sleep times. For each participant, average daily step count, daily time spent stepping, daily time spent standing, and daily time spent sitting/lying were recorded.

### 2.4. ActivPAL Data Processing 

We exported the event-based output from the activPAL software. An event was defined as a single continuous period of a single posture or activity [48]. All events during the self-reported sleeping period (reported in a diary) were excluded (breaking events at the reported sleep times). If not reported, apparent sleep/wake times were estimated based on visual scanning of the data for cessation/resumption of standing or stepping events preceding/following prolonged periods of sitting or lying. Days were considered valid if wear time comprised ≥80% of diary reported waking hours. If waking hours were not reported, ≥10 h of wear time was considered a valid day. Sitting time accumulation data was derived by identifying all sitting bouts during waking, worn time on valid days, and calculating the cumulative sitting time (in minutes and as a proportion of total sitting time) occurring in bouts of any duration. Outcome measures were calculated from the activPAL monitor using a custom Excel macro as per Dall et al. [49]. 

### 2.5. GET READY Intervention (GR)

Staff champions delivered the intervention after receiving a two-hour training session by the main researcher. Each was given an intervention training manual with detailed information of the duration, attendees, aims, pre-session preparation, and conduct of the session, including the activities and tasks. All the material needed to conduct the intervention (e.g., resident’s record of long-term achievement goal/s, flipcharts on strategies, GR goals table, etc.) were provided and a monthly visit/telephone call was scheduled with a researcher to provide feedback. The GR intervention was delivered one-to-one with the care home resident and a relevant family member (TIDieR checklist can be found in Appendix A). It had three main stages over the 12-week period: (1)The familiarisation stage aimed to build a rapport with the resident and the family member and consisted of two sessions, one in week 1 (50–60 min) and the other in week 3 (30–40 min). The first session was used to set one or two long-term achievement goals to sit less and move more, and to help the resident identify 2–4 strategies that they were willing to undertake to sit less and move more. The second session aimed at reviewing the long-term achievement goals and agreeing on which new short-term GR goals and tips the residents would try over the next few weeks. They also reached an agreement between the resident, the family member, and the staff champion on how and when the resident was willing to receive feedback regarding the completion of the strategies embedded in his/her daily routine.(2)The ramping up stage aimed to review the rapport and reach an achievable consensus with the resident and the family member. It consisted of two sessions, one in week 5 and the other in week 7 (20–30 min each). Sessions 3 and 4 were used to review the strategies decided in the previous meeting and their achievement, as well as reviewing the short-term GR goals and tips. If needed, they were modified and adapted.(3)The maintenance stage aimed at integrating behaviours and included two sessions, one in week 9 and the other at week 12 (20–30 min each). Sessions 5 and 6 were used to understand how the resident was getting on with their short-term GR goals, facilitating some problem-solving discussions. They were also aimed at understanding how the resident was getting on with using their current GR strategies and tip/s, agreeing on maintaining/changing the resident’s short-term goals for the next few weeks, and agreeing on maintaining/adapting the resident’s GR tips to use over the few next weeks.

Residents allocated to the CG continued with their usual daily routines. At the end of the study the staff champions in the control sites were given training on the delivery of the intervention so they could offer the GR intervention to their residents.

### 2.6. Data Analysis

Analyses were conducted using the Statistical Package for the Social Sciences (SPSS) software, version 25.0. Descriptive statistics are presented as means and standard deviations (SD) for normally distributed data, median (95% confidence intervals (CI)) for non-normal continuous data or percentages for categories (Table 1). 

As this is a feasibility study, the use of inferential statistics and effectiveness testing is not recommended due to the small sample size and the preliminary nature of the outcomes measured [50]. To assess the preliminary effects of the GR intervention, means and SD are presented, with the mean change from baseline to post-intervention for each outcome calculated along with 95% confidence intervals. Although the feasibility study is not designed to fully understand the effect of an intervention, we have calculated the effect size (Cohen’s d) of the GR intervention on daily time spent sitting/lying. 

## 3. Results

Three care homes were allocated to the intervention group with 22 residents, and one care home was allocated to the CG with nine residents. Baseline characteristics of both groups were similar in terms of age, sex, marital status, number of chronic conditions, number of current medications, and body mass index. Four participants in the CG and eight in the GR used a wheelchair to move around the care home, and four participants in the CG and 10 in the GR needed a walker. Number of current smokers was higher in the CG and a higher percentage of participants from the CG reported being unable to perform several activities of daily living. There was also a two-hour difference in self-reported sitting time on the weekend between the two groups at baseline, the CG being more sedentary. Baseline characteristics of the participants are presented in Table 1.

### 3.1. Feasibility and Acceptability of the Study

Recruitment and enrolment: The initial recruitment target was 40 care home residents from four care homes (two in Glasgow and two in Barcelona). This sample size was a pragmatic decision to ensure that adequate participants were available to estimate the key parameters; recruitment closed at 31 participants, partly as we could only recruit from four care homes. In Glasgow, 21 residents were invited and 9 accepted to participate (42.86%). Reasons for declining were not being interested (n = 2) and health-related issues (n = 10). In Barcelona, 46 residents were invited and 22 participated (47.83%). Reasons for declining were not being interested (n = 7), health-related issues (n = 15), and n = 2 participants were willing to participate but their close family members found their participation an increased risk for adverse health-related issues.

Retention and attendance rates to the intervention: Most of the GR intervention participants (20 out of 22, 90.91%) received what was deemed the minimum dose of face-to-face sessions, at least five out of six face-to-face sessions. Of the two participants (9.09%) not receiving the minimum dose, one withdrew at an early stage due to a long hospitalization and one participant died during the intervention period. Session 1 was anticipated to last 50 min, and the mean (SD) duration time among the intervention sites was 42.33 (13.52) minutes. Sessions 2–6 were anticipated to last between 20 and 35 min each. The mean (SD) duration time among the intervention sites was 25.41 (7.46) min. All sessions were delivered at the intended period, following the intervention manual. In one participant, session 5 was delivered one week later (week 10) due to unavailability of the family member. Overall, the mean (SD) amount of time the staff champion spent in contact with the resident and family member during the 12-week period was 175.52 (11.35) min (almost 3 h). From the monthly calls and discussions, staff champions found the GR intervention practical as it could be delivered with the available resources, time, and commitment or with some combination thereof, and could be easily adapted as the training manual provided for each session several examples of activities appropriate for different participants’ needs. 

Methods of data collection: All participants could complete the questionnaires and physical performance tests (with assistance of the personal legal representative if necessary), suggesting that they were comprehensible and acceptable for these participants. There were no adverse consequences of activity monitoring. However, one participant declined to wear the activPAL monitor.

Acceptability of the assessments and the GR intervention: Twenty-nine (29 out 31, 93.55%) participants ranked their satisfaction with the assessments (ticked 3–5) and all twenty (20 out of 22, 90.91%) ranked their satisfaction with the intervention as completely satisfied (score of 5). 

### 3.2. Preliminary Effects of the GET READY Intervention

The mean (SD) activPAL worn waking hours were 14.2 (1.80) h for the overall sample. Baseline average of the GR and CG levels of activity show that, after standardising for daily wear time, most waking hours were spent sitting/lying followed by standing. Little time was spent stepping, while mean (SD) daily step count was 1226.5 (1085.6) steps in the GR and 1187.4 (974.3) steps in the CG. Participants in the GR group showed a decrease in daily hours spent sitting/lying and an increase in daily hours stepping. However, daily step count did not increase accordingly (see Table 2). The effect size (d) of the GR intervention on daily time spent sitting/lying was 0.36.

The mean change (95% CI) from baseline to post-intervention of the GR participants showed greater improvements than those in the CG in health-related quality of life and perceived overall health in the 0–100 visual analogue scale, fear of falling, and habitual gait peed. The SPPB showed worse results in both groups, although the mean decrease (95% CI) from baseline to post-intervention was smaller in the GR group (see Table 2).

### 3.3. Adverse Events

We had a monthly meeting/telephone call with the staff champions of the three intervention sites during the GR period (three contacts with each intervention site). Staff champions reported not having any adverse event due to the intervention. However, out of the 20 participants that received the minimum dose of GR intervention, two (10%) were hospitalized, five (25%) had a fall, and two (10%) had a relevant physical decline. These events were not associated with the intervention.

## 4. Discussion

### 4.1. Main Findings of This Study 

Our study suggests that a co-created intervention with care home residents, staff, family members, and policymakers is feasible and acceptable to be conducted with care home residents, with the involvement of staff champions and family members. The GR intervention could be fully implemented as planned and proposed, without significant adverse events. The effect size of 0.36 suggests there is potential for the GR study to be effective, but this needs to be confirmed in a future definitive RCT. Improvements in self-rated health and health-related quality of life, fear of falling, and habitual gait speed were also noted in the GR participants.

Even though the GR intervention was shown to be feasible and highly acceptable, we found several challenges engaging care homes in the research process. To understand the care home staff champions’ thoughts on barriers, challenges, facilitators, and key aspects of engaging in research studies, five interviews were performed and the results have been published elsewhere [36]. Even though almost 50% of the residents that were invited by the staff champions participated, which might seem a quite high percentage, the staff champions probably invited the residents most likely to accept and didn’t try to invite residents that do not tend to engage in proposed activities, similar to the case described as the inverse-care law [51]. This is a common issue when offering interventions to older adults within the public health system, which could be overcome by engaging health staff and relevant relatives to help encourage people with the most need. 

It has been recommended that there should be eight general areas of focus addressed by feasibility studies including acceptability, demand, implementation, practicality, adaptation, integration, expansion, and limited-efficacy testing [52]. Staff champions found the GR intervention practical as it could be delivered with the available resources, time, and commitment or with some combination thereof, and could be easily adapted as the training manual provided for each session several examples of activities appropriate for different participants’ needs. To spend three hours during 12 weeks delivering a face-to-face behavioural intervention seemed to be feasible and acceptable for both residents and staff champions to undertake, providing several prompts throughout the day in addition to the face-to-face sessions. Staff champions were receptive to the training manual and found the contents relevant to their work, as noted in the monthly visit/telephone call. Due to high turnover in care home settings, staff champions could be in charge of delivering the training to their peers under supervision. Key characteristics in feasible non-pharmacological interventions include enhancing knowledge and skills of care home staff [53], engaging with families [54], decision-support tools [54], and integration with the care home routines [55]. In addition, research has highlighted the importance of the participatory development of interventions with care home staff, residents, and families [37,54]. The aforementioned characteristics were taken into account alongside the design of the GR intervention.

Even though this pilot study did not aim to evaluate the effects of GR, there were some interesting findings within the outcome measures. GR participants had greater improvements than those in the CG in self-rated health and health-related quality of life. This result could have two possible explanations. The first emphasizes the well-known perceived health-related benefits of sitting less and moving more and, the second one could be related to a better health perception due to receiving more attention from the staff champions and the family member who engaged. Health-related quality of life is part of a multidimensional approach that considers physical, mental, and social aspects [56]. Increases in PA have been shown to be associated with better self-rated health and health-related quality of life in numerous studies [56,57], related to an improved mood and an enhanced sense of well-being [58]. Even quite modest improvements in mental and physical health are likely to produce large relative increases in the number of quality-adjusted life-years to care home residents, who have an extremely poor baseline level of health and life expectancy [59]. 

Moving to a care home may influence the risk of experiencing loneliness and loss as a potentially negative aspect of lacking (diminishing) social relations [60]. A previous study showed that care home residents’ perceived satisfaction would positively affect sense of belonging and would be negatively associated with loneliness, thus playing a protective role against the experience of loneliness. More regular social interactions with the staff champions and a family member, as well as receiving regular prompts from members of staff as part of the GR intervention might have enhanced the participants’ sense of belonging, accompanied by perceiving a better health status and health-related quality of life. A recent systematic review showed that interventions that enhanced social connections and support improved social outcomes and health-related quality of life [61]. In addition, some of the studies included in the review found that the care home staff implementing the intervention also experienced benefits from it such as improved communication with the residents [61]. These findings point towards an additional need to focus on the staff involved when implementing interventions. If such interventions are to be successfully and sustainably delivered, they need to be embedded in routine practice, and care home staff should be involved in developing and delivering the necessary change [62]. 

GR participants also increased habitual gait speed, which might be linked to a decrease in daily hours spent sitting/lying and an increase in daily hours stepping. PA plays an important role in improving gait speed in older adults [63], and thus increased movement may give the resident more confidence in their walking ability. Recommended criteria for clinically meaningful change when measuring the habitual walking speed of community dwelling older adults measured over 4 or 10 m is 0.1 m/s for substantial meaningful change [64]. The mean change in gait speed in our participants from the GR was 0.2 m/s. Baseline levels of time spent walking and step count in both CG and GR residents were low, but were similar to those of older adults in an Australian care home [65]. Accelerometers and pedometers generally become less accurate as walking speed decreases [66,67]. The activPAL tends to underreport step count at gait speeds less than 0.47 m/s [68]. Our CG and GR participants had a mean baseline gait speeds of 0.3 and 0.4 m/s, respectively, so the activPAL could have underestimated their daily number of steps. The time spent walking by the GR residents increased by about 20 min a day with the intervention, but the number of daily steps taken did not increase accordingly. This is unexpected, but may be because the measurement of steps from sporadic daily activities is less accurate than for continuous periods of purposeful walking. In activPAL validation studies, step detection was less accurate (under-reported) during simulated activities of daily living than during continuous walking [46,69], whilst the duration of walking was equally accurate for both types of movement [46]. It is therefore likely that the GR residents did increase their duration of walking, but in a slow, shuffling, and sporadic manner so that the additional steps were not recorded.

### 4.2. What Is Already Known on This Topic

The average daily hours spent sitting/lying in our sample are in line with those observed in similar populations [65]. Care-home residents are among the most sedentary and less active of the population. It is well known that individuals who enter residential aged care are often suffering from debilitating health problems that impair their general mobility. However, it is commonly found in long-term settings that individuals can move independently, but instead are being helped beyond their abilities [70]. Excess disability could be reduced by increasing opportunities for independent activity [70], even in severely cognitively impaired and functionally disabled nursing home residents [71]. Thus, strategies to enhance movement are in line with public health priorities. 

### 4.3. What This Study Adds

To our knowledge, this is the first study to assess the feasibility, acceptability, and preliminary effects of a co-created intervention to reduce SB and enhance movement throughout the day in care home residents. Co-creation involved end-users, relevant relatives and staff members in co-designing this health-related intervention. Regular prompts for movement in line with goals set with the resident appears feasible and acceptable. 

### 4.4. Limitations of This Study

There are some limitations to the present study. Even though the main aim of the pilot study was not to evaluate the effects of the GR intervention, post-intervention assessments were not blinded to the site allocation, implying a possible source of bias. Also, there was an imbalanced number of participants in the CG due to the withdrawal of a care home allocated to the CG which could have altered the results. The CG care home had more residents who smoked, had lower functional ability, and sat more at weekends than the intervention group homes. It would be important for a future study that assesses the effects of an intervention to attempt to recruit homes with similar residents so that the randomisation allows similar group baseline characteristics. However, this is a challenge that has been commonly reported when doing research studies in care home settings. Finally, we assessed attendance to the face-to-face sessions, but adherence to the behaviour change strategies (e.g., goal-setting) and prompts were not collected. Due to time constraints in a busy care home environment, this information would have been inaccurate. 

## Figures and Tables

**Table 1 ijerph-17-02866-t001:** Baseline characteristics of the care home residents participating in the GET READY study (n = 31).

Characteristics	CG (n = 9)	GR (n = 22)	Total (n = 31)
Women, n (%)	6 (66.7)	10 (45.5)	16 (51.6)
Age, mean (SD)	82.7 (13.2)	83.2 (9.1)	82.9 (13.6)
Marital status - single or widow, n (%)	7 (77.8)	18 (81.1)	25 (80.6)
Number of medical conditions, mean (SD)	6.7 (1.7)	6.8 (2.4)	6.8 (2.2)
Number of current medications, mean (range)	10.1 (2.7)	9.8 (3.4)	9.9 (3.2)
BMI (kg/m^2^), mean (SD)	25.3 (5.5)	27.3 (5.5)	26.7 (5.5)
Tobacco users, n (%)	6 (66.7)	7 (31.8)	13 (41.9)
Katz Index, n that said “not able” (%)			
Bathing	9 (100)	18 (81.8)	27(87.1)
Dressing	6 (66.7)	12 (54.5)	18 (58.1)
Toileting	8 (88.9)	17 (77.3)	25 (80.6)
Transferring	8 (88.9)	17 (77.3)	25 (80.6)
Continence	6 (66.7)	15 (68.2)	21 (67.7)
Feeding	1 (11.1)	2 (9.1)	3 (9.7)

CG: Control Group; GR: Get Ready group; SD: Standard Deviation; BMI: Body Mass Index.

**Table 2 ijerph-17-02866-t002:** Outcomes at baseline and post intervention with mean change between time points.

	GR (n = 22)			CG (n = 9)		
Outcome	Baseline	Post	Mean Change (95% CI)	Baseline	Post	Mean Change (95% CI)
SBQ						
Hours sitting on a week day, mean (SD)	8.6 (2.8)	8.2 (3.9)	−0.4 (−3.6, 3.9)	8.9 (3)	8.8 (3.7)	−0.1 (−0.9, 2.4)
Hours sitting on a weekend day, mean (SD)	9.0 (2.6)	8.7 (2.9)	−0.3 (−3.3, 3.4)	7.4 (3.1)	7.7 (3.4)	0.3 (−1.7, 2.8)
Sedentary behaviour						
Daily step count, mean (SD)	1226.5 (1085.6)	1249.4 (906.6)	22.9 (−131.5, 254.3)	1187.4 (974.3)	971.7 (804.1)	−215.7 (−312, 295.5)
Daily time sitting/lying (h), mean (SD)	12.7 (3.6)	11.9 (2.8)	−0.8 (−1.9, 1.8)	12.9 (2.8)	12.7 (1.7)	−0.2 (−1.2, 1.4)
Daily time standing (h), mean (SD)	1.7 (1.4)	1.4 (1.4)	−0.3 (−1.9, 2.1)	1.4 (1.9)	1.3 (0.8)	−0.1 (−2.1, 1.6)
Daily time stepping (minutes), mean (SD)	34.5 (23.2)	58.9 (36.7)	24.4 (−16.2, 48.1)	31.2 (28.5)	33.6 (29.1)	2.4 (−9.2, 10.5)
Self-rated health-related quality of life						
EQ-5D score (out of 15 points *), median (IQR)	10 (7, 15)	8 (7, 13)	−2 (−4, 6)	11 (8, 15)	12 (8, 15)	1 (−3, 5)
EQ-VAS (0-100 scale), mean (SD)	53.9 (19.5)	61.4 (14.3)	7.5 (−24.7, 17.5)	54.8 (21.6)	56.8 (19.2)	2 (−31.4, 14.3)
Fear of falling MFES (out of 140 points **), mean (SD)	95 (31)	111 (38)	16 (−20, 13)	99 (35)	97 (28)	−2 (−19, 16)
Functional performance						
SPPB (out of 12 points ***), mean (SD)	5.4 (2.7)	5.2 (2.8)	−0.2 (−1.1, 3.2)	5.5 (3.1)	4.9 (2.6)	−0.6 (−1.7, 1.8)
Habitual gait speed (m/s), mean (SD)	0.4 (0.4)	0.6 (0.9)	0.2 (−0.2, 0.4)	0.3 (0.6)	0.3 (0.7)	0 (−0.1, 0.3)

* Higher scores indicate worse perceived quality of life. ** Higher scores indicate better confidence undertaking certain tasks (e.g., get in/out of chair). *** Higher scores indicate better physical function. GR: Get Ready group; CG: Control Group; SD: Standard Deviation; CI: Confidence Interval; SBQ: Sedentary Behaviour Questionnaire; EQ-5D: Europe Quality of Life; VAS: Visual Analogue Scale; MFES: Modified Falls Efficacy Scale; SPPB: Short Physical Performance Battery.

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
