# Peer review of "A Pilot Randomised Clinical Trial of a Novel Approach to Reduce Sedentary Behaviour in Care Home Residents: Feasibility and Preliminary Effects of the GET READY Study"

_ijerph, 2020, doi:10.3390/ijerph17082866_

Round 1

Reviewer 1 Report

A pilot randomised clinical trial of a novel approach to reduce sedentary behaviour in care home residents:  feasibility and preliminary effects of the GET READY study

In the present study, authors present the preliminary results of the Get Ready study, more precisely they aimed to assess the feasibility, acceptability, safety and preliminary effects from that research. Even the main aim of the pilot study was not to evaluate the effects of the GR intervention, the preliminary effects were presented.

The study is extremely relevant given the increase in life expectancy of the population and in particular the increase in the population living in nursing homes.

The introduction is well presented and allows the reader to understand the study problem. The methods are well described.

After reading the article, emerged me some questions that I would like to see clarified.

A first question concerns the sample size. Some calculation has been done to determine what sample size is needed to be able to draw conclusions about the feasibility, acceptability, safety of the present study? On the official RCT page (https://clinicaltrials.gov/ct2/show/record/NCT03505385), the authors point to the need to integrate 60 subjects. Where were based for the determination of this number?

A second issue is related to the need to perform or not, statistical analysis of the data collected. There seems to be some contradiction throughout the article. Firstly, on data analysis the authors refer “As this is a feasibility study, the use of inferential statistics and effectiveness testing is not recommended due to the small sample size and the preliminary nature of the outcomes measured.” If it´s only a feasibility study, it is not clear to me, initially, what is the need for a control group, also the need to present results from the beginning and end of the study in the various variables, and finally why did they only study the effect size? In the same section of the data analysis, the authors mention that “Although the feasibility study is not designed to fully understand the effect of an intervention, we have calculated the effect size (Cohen’s d) of the GR intervention on daily time spent sitting/lying to allow calculation of a sample size for a future definitive clinical trial.” If what is written is correct, and since it does not contribute to the objectives of this study (the use of the effect size was made only to calculation of a sample size for the future clinical trial), I suggest removing it from the text, as it will only have to be described in a future article on the same study. In the study mentioned by the authors (Abbott 2014) to justify most of the procedures, it highlights at a given moment that “Considerable caution is advised in the use of pilot study outcomes data to generate sample size calculations, as the estimates may be biased due to factors that may not be reproduced in a full trial, or may be unrealistic due to chance factors related to the small sample size”. Secondly, in the title of the article, it mentions that it is intended to evaluate “feasibility and preliminary effects”. I believe that given the small sample size, there is a need to be careful with the generalization of the results (with due emphasis on this fact in the limitations of the study), but on the other hand, treating the data only in a descriptive way as performed here, also seems to me not to be the best solution. Same study mentioned (Abbott 2014) refers that “They may include within-group treatment effects, such as effect size, but these should not be the basis of sample-size calculations unless the MID for the outcome measure is genuinely not known.” So, if the authors want to present preliminary effects why didn’t present also the effect size on the other variables?

Could the fact that data were collected in 3 institutions in the intervention group and only one institution in the control group, may have influenced the results? Moreover, because there seems to be clear differences between groups (although in table 1 does not introduce any statistic excepted the descriptive ones), in particular on the number of subjects, the percentage of women and the percentage of individuals who assume that they are unable to fulfill the tasks mentioned in the Katz Index.

About the feasibility and acceptability of the study, participation in the study among those invited in Glasgow was 42.86%. What was the percentage in Barcelona? Was it different? Greater or lesser adhesion?

Minor revisions:

The information in the text on the lines 245-6 “(n=1 participant from the 245 GR group died during the study period and n=1 from the GR group withdrew at an early stage)”  had already been mentioned in lines 225-6

Author Response

We would like to thank the reviewers for their time and effort and have made our responses to each Reviewer below.

Reviewer #1

A first question concerns the sample size. Some calculation has been done to determine what sample size is needed to be able to draw conclusions about the feasibility, acceptability, safety of the present study? On the official RCT page (https://clinicaltrials.gov/ct2/show/record/NCT03505385), the authors point to the need to integrate 60 subjects. Where were based for the determination of this number?

The authors appreciate your comment and we updated the protocol at clinicaltrials.gov on January 23rd 2020 with the actual number of participants enrolled. Since we aimed to conduct a feasibility study, we reviewed the literature to determine the best approach to calculate an optimal sample size. We found a useful introduction to feasibility and pilot studies given by Lancaster, Dodd & Williamson (2004) who state that the sample size needs to be calculated according to our aim:

  • If we want to estimate a parameter such as a standard deviation which will be used in a sample size calculation for the full-scale trial, they recommend an overall sample size of 30 (rule of thumb).
  • If we want to estimate the rate (proportion) of eligible people who are willing to participate, of participants who drop out of the trial, or of participants who comply with their allocated intervention, then a simple approach is to relate the proposed sample size to the width of the 95% confidence interval for the rate. For example, you might say “with a sample size of 50, we will be able to estimate a drop-out rate of 80% to within a 95% confidence interval of +/- 11%”.

The authors aimed a sample size of 60 participants before the beginning of the study because our main aim was the second option. With the withdrawal of one care home, and the reduced recruitment time, we needed to start the feasibility study with 31 participants. We are unsure if it helps to add text to the manuscript explaining this level of detail so should you require this information we will add it. As yet we have made no changes to the manuscript.

Reference:

Lancaster GA, Dodd S, Williamson PR. Design and analysis of pilot studies: recommendations for good practice. J Eval Clin Practice 2004;10:307-312.

A second issue is related to the need to perform or not, statistical analysis of the data collected. There seems to be some contradiction throughout the article. Firstly, on data analysis the authors refer “As this is a feasibility study, the use of inferential statistics and effectiveness testing is not recommended due to the small sample size and the preliminary nature of the outcomes measured.” If it´s only a feasibility study, it is not clear to me, initially, what is the need for a control group, also the need to present results from the beginning and end of the study in the various variables, and finally why did they only study the effect size? In the same section of the data analysis, the authors mention that “Although the feasibility study is not designed to fully understand the effect of an intervention, we have calculated the effect size (Cohen’s d) of the GR intervention on daily time spent sitting/lying to allow calculation of a sample size for a future definitive clinical trial.” If what is written is correct, and since it does not contribute to the objectives of this study (the use of the effect size was made only to calculation of a sample size for the future clinical trial), I suggest removing it from the text, as it will only have to be described in a future article on the same study. In the study mentioned by the authors (Abbott 2014) to justify most of the procedures, it highlights at a given moment that “Considerable caution is advised in the use of pilot study outcomes data to generate sample size calculations, as the estimates may be biased due to factors that may not be reproduced in a full trial, or may be unrealistic due to chance factors related to the small sample size”. Secondly, in the title of the article, it mentions that it is intended to evaluate “feasibility and preliminary effects”. I believe that given the small sample size, there is a need to be careful with the generalization of the results (with due emphasis on this fact in the limitations of the study), but on the other hand, treating the data only in a descriptive way as performed here, also seems to me not to be the best solution. Same study mentioned (Abbott 2014) refers that “They may include within-group treatment effects, such as effect size, but these should not be the basis of sample-size calculations unless the MID for the outcome measure is genuinely not known.” So, if the authors want to present preliminary effects why didn’t present also the effect size on the other variables?

The authors thank you for your comments and have tried to clarify our intentions.

  • First of all, we have tried to clarify (Page 4, lines 103-107) that our main goal was to assess the feasibility, acceptability and safety of the Get Ready intervention. A secondary aim was to assess the preliminary effects of the intervention to reduce sedentary behaviour and improve health-related outcomes.
  • Feasibility studies can be conducted prior to a definitive RCT, to improve the chances of success. Our reasons for conducting a feasibility study were: to inform process (e.g., feasibility of recruitment, retention, intervention adherence), to understand resource requirements (e.g., time and budget issues), to inform management (e.g., personnel challenges, data collection or organization), and to advance scientific inquiry (e.g., intervention safety, appropriate dose, potential treatment effect). In order to mimic the structure of a future RCT, collect information about the usual care and existing interventions to increase movement and reduce sedentary behaviour, and collect the feelings of staff champions, we considered it relevant to have a control group, mostly to assess the aforementioned feasibility variables, rather than comparing the effects between groups.
  • We agree that the calculation of the effect size on daily time spent sitting/lying was not the main aim of our study. However, we thought it could be informative for the reader as a preliminary result. We decided to give only the effect size of the main outcome since the intervention is aimed at reducing sedentary behaviour. Should the reviewer consider it unnecessary, we would delete it.
  • We appreciate the reviewers’ comments and Abbott’s paper (2014) regarding the limitations of using feasibility studies to calculate sample size for a future RCT. We feel it is appropriate to still add the effect size but have removed mention of the use of this for sample size for a future study (Page 10, line 255).

Could the fact that data were collected in 3 institutions in the intervention group and only one institution in the control group, may have influenced the results? Moreover, because there seems to be clear differences between groups (although in table 1 does not introduce any statistic excepted the descriptive ones), in particular on the number of subjects, the percentage of women and the percentage of individuals who assume that they are unable to fulfill the tasks mentioned in the Katz Index.

The authors agree with your comment and this situation was added as a limitation but has now been made more explicit (Page 17, lines 447-449). As mentioned earlier in the manuscript, we aimed to have another control care home but it had to be withdrawn due to lack of fluent communication. Engaging care homes and other long-term care institutions to research projects is a challenge that had been commonly reported in several studies. In future studies, we will need to consider recruiting more care homes in case one setting needs to be withdrawn.

About the feasibility and acceptability of the study, participation in the study among those invited in Glasgow was 42.86%. What was the percentage in Barcelona? Was it different? Greater or lesser adhesion?

The authors appreciate your comment and apologise for not including this information. The following sentence has been added in the results – recruitment and enrolment section (Pages 10-11, lines 273-279):

“In Glasgow, 21 residents were invited and 9 accepted to participate (42.86%). Reasons for declining were not being interested (n=2) and health-related issues (n=10). In Barcelona, 46 residents were invited and 22 participated (47.83%). Reasons for declining were not being interested (n=7), health-related issues (n=15), and n=2 participants were willing to participate but their close family members found their participation an increased risk for adverse health-related issues”.          

The information in the text on the lines 245-6 “(n=1 participant from the GR group died during the study period and n=1 from the GR group withdrew at an early stage)” had already been mentioned in lines 225-6.

The authors appreciate your comment and have deleted the later sentence.

Reviewer 2 Report

I congratulate the authors of the article. The article is very well written, with academic and scientific quality. In addition, he advanced in the gerontological area by proposing intervention for a group of older people, with the outcome of sedentary behavior. As you will see in my comments in the attached file, the introduction needs to be revised, as it does not present a gap in the literature to be filled (and it exists), nor does it present a justification for the study, much less the hypotheses of results.

Author Response

We would like to thank the reviewers for their time and effort and have made our responses to each Reviewer below.

Reviewer #2

Abstract:

The authors had briefly added the data analysis, and have moved the participants’ information to the methods section, as requested.

Introduction:

Which is the justification for conducting the study? Which is the gap in the gerontological literature you want to fill? What are the hypotheses of results?

The authors have tried to make the justification of the study clearer to the reader and have underlined in yellow the information you are asking for. We have also added a hypothesis.

Results:

We have moved the participants’ information to the methods section, as requested.

Reviewer 3 Report

I think this was a well-written manuscript, presenting the preliminary evaluation of the GET READY intervention that potentially changes sedentary behaviors of care home residents. I found that the evaluation process and the measurement instrument selection were appropriate. However, I expect to learn the overall picture of the whole study without referring to other papers, resources, or information from the ClinicalTrial website. Therefore, several concepts need to be further explained, especially for international readers, and some issues need to be justified:

  • Provide clear definition of Care Home. What is care home in Scotland and Spain? What kind of services it provides to older people? How does it look like? How many residents does it usually accommodate? Are they different in these two countries?
  • Why “cluster randomized control trial” was used? Why not RCT? As the intervention was administered to individual residents, the effect on other residents—in the same facility—who did not receive the intervention would be minimal. Having both case and control subjects within one care home would provide stronger, more rigorous research design.
  • I am not convinced having one control site in Glasgow was appropriate, although the intervention and control sites were randomly assigned. The advantages of having a control group, as we all know, are to rule out confounding factors such as public policies, national/international event/incidence (e.g., COVID-19), and others. I am not sure how a Glascow site could serve as a control if, for example, the behaviors of residents in Barcelona changed due to a recent national incentive policy issued by the Spain government during the study period. In this case, this was a very weak control.
  • How were the care homes selected? Was it a convenient sample? What were the inclusion and/or exclusion criteria? Why in two different countries? Would cultural difference play an important role in this study? I hope to get this information without reading other papers.
  • For the subjects, I would like to see information such as the ability to walk and whether they walk with assistive devices (cane, walker and wheelchair), especially for studies of this kind related to physical activities and sedentary behaviors.
  • Should “being able to walk” be one of the subject inclusion criteria?
  • Page 8, 4th paragraph, described the Staff Champion’s comments on the process. It was not mentioned how the information was collected in the method section, nor presented in the results.

Minor issues:

  • Abstract: Please spell out CG and RCT.
  • Table 1 is at the wrong place; should be on page 5, after line 214
  • Please introduce the GR intervention (page 4-5, Line 158-190) at the beginning of the manuscript, as part of the study background. This information should not be in the method section.

Author Response

We would like to thank the reviewers for their time and effort and have made our responses to each Reviewer below.

Reviewer #3

Provide clear definition of Care Home. What is care home in Scotland and Spain? What kind of services it provides to older people? How does it look like? How many residents does it usually accommodate? Are they different in these two countries?

The authors agree this is an interesting point. In a previous stage of the GET READY study we conducted discussion groups in care homes from both countries and could verify that the definition of a care home is similar in terms of services provided, ratio of staff per residents, and skills of staff members. In terms of size (number of beds), there are different sizes in both countries so that we selected one big care home (≥100 beds) and one medium-sized (40-99 beds) in each country. We have added the following definition in the participants section (Page 5, lines 131-134): “A care home is a long-term care setting where people live and have their care needs met in homely surroundings, usually for people needing more care than they could get in their own home or in supported housing”.

Reference:

Giné-Garriga M, Sandlund M, Dall P, Chastin S, Pérez S, Skelton D. A Novel Approach to Reduce Sedentary Behaviour in Care Home Residents: The GET READY Study Utilising Service-Learning and Co-Creation. Int J Environ Res Public Health. 2019;16(3):418.

Why “cluster randomized control trial” was used? Why not RCT? As the intervention was administered to individual residents, the effect on other residents—in the same facility—who did not receive the intervention would be minimal. Having both case and control subjects within one care home would provide stronger, more rigorous research design.

The authors understand your comment but our strongest concern with conducting an RCT was the difficulty of staff champions in charge of conducting the intervention to distinguish between messages given to control and intervention participants. Our aim was to train staff members to be able to conduct the intervention to enhance long-term implementation and sustainability. If we had trained all staff champions with the complete training manual it would have been difficult for them to provide some prompts and strategies only to some residents and keep their usual care to the control participants (since staff members would have further knowledge). It would have also been hard to know how much contamination had occurred in each setting since we were not physically conducting the intervention (but only doing periodic follow-ups).  The majority of care home exercise/movement studies are conducted as cluster randomised trials to avoid potential contamination.

I am not convinced having one control site in Glasgow was appropriate, although the intervention and control sites were randomly assigned. The advantages of having a control group, as we all know, are to rule out confounding factors such as public policies, national/international event/incidence (e.g., COVID-19), and others. I am not sure how a Glasgow site could serve as a control if, for example, the behaviors of residents in Barcelona changed due to a recent national incentive policy issued by the Spain government during the study period. In this case, this was a very weak control.

There was only one control site in Barcelona, we planned to have another control site in Glasgow but as mentioned earlier in the manuscript, it had to be withdrawn due to lack of fluent communication. The authors understand your concern and agree with your comment, and this situation was added as a limitation. Engaging care homes and other long-term care institutions to research projects is a challenge that had been commonly reported in several studies. In future studies, we will need to consider recruiting more care homes in case one setting needs to be withdrawn.

How were the care homes selected? Was it a convenient sample? What were the inclusion and/or exclusion criteria? Why in two different countries? Would cultural difference play an important role in this study? I hope to get this information without reading other papers.

It was a convenience sample. The inclusion criteria were to be public care homes with similar sizes in both countries. We selected one big care home (≥100 beds) and one medium-sized (40-99 beds) in each country. Analysing the cultural differences between Glasgow and Barcelona wasn’t within the scope of the present study, but we thought it would offer an added value conducting the study in both countries. We have added this information to the methods (page 5 lines 133-134). As mentioned earlier, in a previous stage of the GET READY study we conducted discussion groups in care homes from both countries and could acknowledge some cultural differences mainly related to the types of movement-based activities and sedentary behaviour patterns. However, as the intervention was meant to have a similar structure but be individualized to and co-designed with each participant, cultural differences may be interesting in terms of analysing the effects rather than conducting a feasibility and acceptability study.

For the subjects, I would like to see information such as the ability to walk and whether they walk with assistive devices (cane, walker and wheelchair), especially for studies of this kind related to physical activities and sedentary behaviors. Should “being able to walk” be one of the subject inclusion criteria?

The authors have added more information regarding their ability to walk in the results section, as requested (page 10, lines 261-263). Care home residents tend to have mobility restrictions and a high percentage are not able to walk independently. As we aimed to individualize the strategies to reduce sedentary behaviour and increase movement throughout the day, and target as many residents as possible, we did not want to limit the inclusion criteria. It is commonly found in long-term settings that some individuals can move independently but instead are using a wheelchair and being helped despite their abilities. In these cases, we may have margin for improvement. When there is further mobility limitations, other strategies such us assisted standing several times a day might be a feasible strategy. That is why staff members and family members needed to be engaged.

Page 8, 4th paragraph, described the Staff Champion’s comments on the process. It was not mentioned how the information was collected in the method section, nor presented in the results.

The authors thank you for your comment and have added that we collected information about adverse events (e.g. safety issues) and other intervention-related issues in a monthly meeting/ telephone call with the Staff Champions (page 7 lines 167-168). We have also clarified some information in the results-adverse events section, as requested (page 12 lines 325-326).

Abstract: Please spell out CG and RCT.

The authors have named the control group (CG) and the Get Ready intervention (GR) in the abstract and throughout the document.

Table 1 is at the wrong place; should be on page 5, after line 214.

We are afraid we don’t understand your comment. The authors’ guidelines of the journal state that all tables should be located after the references.

Please introduce the GR intervention (page 4-5, Line 158-190) at the beginning of the manuscript, as part of the study background. This information should not be in the method section.

The authors appreciate your comment and had already introduced the co-design of the GR intervention in the background section, and had added a reference. We believe that having some information in the methods section about the main characteristics of the GR intervention is important in order to understand how it was conducted. We believe that the readers need to know about the stages of the intervention and the time and resources required at each stage. 

Round 2

Reviewer 3 Report

The authors have sufficiently addressed my previous concerns and I recommend this manuscript to be published.